# Progress on Studies of Beams Carrying Twist

**Zhenglin Liu****, Lipeng Wan, Yujie Zhou, Yao Zhang and Daomu Zhao ***

Zhejiang Province Key Laboratory of Quantum Technology and Device, Department of Physics, Zhejiang University, Hangzhou 310027, China; zhenglinliu@zju.edu.cn (Z.L.); 11736039@zju.edu.cn (L.W.); 21736051@zju.edu.cn (Y.Z.); 21836051@zju.edu.cn (Y.Z.)

\* Correspondence: optics@zju.edu.cn

**Abstract:** Optical twist has always been a hot spot in optics since it was discovered in 1993. Twisted beams can be generated by introducing the twist phase into partially coherent beams, or by introducing the twisting phase into anisotropic beams, whose spectral density and degree of coherence will spontaneously rotate during propagation. Unlike conventional beams, twisted beams have unique properties and can be used in many applications, such as optical communications, laser material processing, and particle manipulation. In this paper, we present a review of recent developments on phase studies of beams carrying twist.

**Keywords:** twist phase; twisting phase; transmission characteristics; rotation

## 1. Introduction

Twist phase was first discovered by Simon and Mukunda in 1993 when they were looking for the most general rotationally invariant partially coherent light field [1]. Different from ordinary phase, its internal asymmetry causes the light spot to rotate during transmission. Due to the positive definite condition, the twist phase is restricted by the coherence of the light field and can only exist in partially coherent fields [1]. Previously, the twisted partially coherent beams were mainly processed by the method of Wigner distribution function, the representative of which was the twisted Gaussian Schell-model (TGSM) beam [2]. Sundar et al. analyzed the propagation characteristics of TGSM beams, and studied the influence of twist phase on the beam focus and focus shift [3–6]. TGSM beams were discovered to be produced by incoherent superposition of ordinary Gaussian beams, laying the foundation for the experimental generation [7,8]. After that, the method of mode decomposition and tensor analysis were proposed and applied to analyze the light field carrying the twist phase, which promoted the studies of analysis of beam properties, propagation, and transformation [9,10]. Simon and Mukunda studied anisotropic TGSM beams under the condition of the full ten-parameter [11] and TGSM solitons were proved to exist in a special nonlinear medium [12]. The twisted phase is found to be closely related to the vortex phase. The TGSM beam could be regarded as a combination of infinite uncorrelated spiral components, or as an incoherent superposition of partially coherent modified Bessel–Gaussian beams [13,14]. The internal asymmetry of the twist phase makes the beam exhibit unusual orbital angular momentum (OAM) properties. The relationship between the twist strength coefficient and the OAM has been deeply studied, and it is found that the distribution of the OAM of the beam carrying the twist phase behaves like a rotating rigid body [15–17].

Recently, the studies on twisted beams are becoming more and more diversified. Borghi et al. solved the problem about the feasibility of the existence of the twist phase in any partially coherent fields, and proposed a necessary and sufficient condition for mapping typical Schell-model partially coherent cross-spectral density (CSD) functions to truly twisted CSD functions [18,19]. Gori and Santarsiero introduced a modeling procedure that can be used to generate genuine twisted sources without symmetry constraints [20].

With the development of beam design, a large number of new partially coherent beams carrying the twist phase have been studied [21–29]. Wu introduced a radially polarized partially coherent twisted beam [21]; Mei and Korotkova proposed the flat-top twisted rectangular multi-Gaussian Schell-model source, the electromagnetic (EM) TGSM source and the EM twisted multi-Gaussian Schell-model beam [10,22]; Wan and Zhao introduced the twisted Gaussian Schell-model array beam, and a family of partially coherent beams with high-order twist phase [23,24]; the twisted Laguerre Gaussian-Schell model beam was proposed by Peng et al. [25]; the EM twisted Gaussian Schell-model array beam was studied by Zhou and Zhao [26]; Luo and Zhao studied the anisotropic twisted Laguerre Gaussian-Schell model and the EM twisted Laguerre Gaussian-Schell model beams in detail [27,28]; Zheng et al. proposed the ring-shaped twisted Gaussian Schell-model array beam [29]; Hyde studied the twisted space-frequency and space-time partially coherent beams [30]. The experimental generation of twisted beams has also made great progress. Wang et al. produced the true TGSM beam by a system including three cylindrical mirrors [31], whereas Tian et al. demonstrated how to generate twisted Schell-model beams by realizing the continuous coherent beam integral in discrete form [32].

The applications of twisted beams are mainly reflected in the propagation in turbulence. Researchers have clearly studied the transmission characteristics and special properties of TGSM beams, twisted anisotropic Gaussian Schell-model beams, twisted rectangular multi-Gaussian Schell-model beams propagating, etc., in atmospheric turbulence, oceanic turbulence, weak turbulence, and anisotropic non Kolmogorov turbulence, which has broad application prospects in optical communication [33–38]. In other fields, twisted beams have also been fully studied, such as ghost imaging, particle manipulation, image construction, and self-reconstruction [39–42].

Different from the traditional twist phase, our group proposed a kind of twisting phase, which can be regarded as the extension of the astigmatic phase [43]. The study of the astigmatic phase first appeared in the mode converter, which can realize the conversion of Hermite-Gaussian beam and Laguerre-Gaussian beam [44,45]. The orbital angular momentum of the beam obtained by astigmatism transformation is studied by Courtial et al. [46]. Kotlayer studied the properties of beams carrying astigmatic phase [47,48]. In addition, Ponomarenko also explored the separable phase carrying orbital angular momentum [49]. It was shown that the beam with twisting phase exhibits a twist structure similar to TGSM beams [50,51], even its spectral density and degree of coherence (DOC) could break through the limitation of the angle and direction of rotation [43,52]. The effect of the twisting phase in particle manipulation was studied by Zhang et al. [53].

In this article, we will briefly introduce the basic expressions of the traditional twist phase and the twisting phase, and introduce the similarities and differences between them. Then, we will introduce recent developments on the beams carrying on the traditional twist phase or twisting phase.

## 2. The Twist Phase and the Twisting Phase

Twist phase is a non-separable quadratic phase that could cause the rotation of the beam spot during transmission. Simon and Mukunda proposed a class of sources called TGSM sources where the twist phase could be imposed on the conventional Gaussian Schell-model beams. The spatial coherence properties of partially coherent beams could be described by the CSD function in the space-frequency domain [54]. In order to achieve a sufficient nonnegative definiteness condition, the following equation must be satisfied for a nonnegative weight function $p(\nu)$ and an arbitrary kernel function $H_0$ [55]

$$W_0(\boldsymbol{r}_1, \boldsymbol{r}_2) = \int p(\boldsymbol{\nu}) H_0^*(\boldsymbol{r}_1, \boldsymbol{\nu}) H_0(\boldsymbol{r}_2, \boldsymbol{\nu}) d^2\boldsymbol{\nu}. \tag{1}$$

In order to introduce the sources generating the TGSM beam, kernel function is defined as [10]

$$H_0(\mathbf{r}, \mathbf{v}) = \exp(-\sigma \mathbf{r}^2) \exp\{-[(ay + ix)v_x - (ax - iy)v_y]\}, \tag{2}$$

where $\sigma$ and $a$ are positive real constants. The weight function is chosen as a Gaussian profile

$$p(\mathbf{v}) = (\alpha/\pi) \exp[-\alpha(v_x^2 + v_y^2)], \tag{3}$$

where $\alpha$ is a positive real constant and satisfy $u = a/\alpha$.

Substituting Equations (2) and (3) into Equation (1), the CSD function of the traditional TGSM beam is expressed as

$$W_T(\mathbf{r}_1, \mathbf{r}_2) = \exp\left(-\frac{r_1^2 + r_2^2}{4\sigma_0^2}\right) \exp\left[-\frac{(\mathbf{r}_1 - \mathbf{r}_2)^2}{2\delta_\mu^2}\right] \exp[-iu(x_1 y_2 - x_2 y_1)], \tag{4}$$

where $\sigma_0$ is the width of the light spot, $\delta_\mu$ is related to the coherence width, and $u$ characterizes the twist strength. It was further found by Simon and Mukunda that the twist strength should satisfy inequality $u\delta_\mu^2 \leq 1$ to ensure the source is physically realizable, which means that the traditional twisted phase can only exist in partially coherent beams [1].

The traditional twist phase is inseparable. Consider another set of kernel and weight function

$$H_0(r, v) = \tau(r) \exp(-2\pi i v \cdot r) \exp[iu(x\cos\theta - y\sin\theta)(x\sin\theta + y\cos\theta)], \tag{5}$$

$$p(v) = 2\pi\delta_x\delta_y \exp(-2\pi^2\delta_x^2 v_x^2) \exp(-2\pi^2\delta_y^2 v_y^2). \tag{6}$$

Substituting Equations (5) and (6) into Equation (1), the CSD function is calculated as

$$\begin{aligned} W_N(\mathbf{r}_1, \mathbf{r}_2) = {} & \exp\left(-\frac{x_1^2 + x_2^2}{4\sigma_x^2}\right) \exp\left(-\frac{y_1^2 + y_2^2}{4\sigma_y^2}\right) \exp\left[-\frac{(x_1 - x_2)^2}{2\delta_x^2}\right] \exp\left[-\frac{(y_1 - y_2)^2}{2\delta_y^2}\right] \\ & \times \exp\{-iu[(x_1\cos\theta - y_1\sin\theta)(x_1\sin\theta + y_1\cos\theta) \\ & - (x_2\cos\theta - y_2\sin\theta)(x_2\sin\theta + y_2\cos\theta)]\}. \end{aligned} \tag{7}$$

where $\sigma_x$, $\sigma_y$, $\delta_x$ and $\delta_y$ are the transverse intensity widths and spatial coherence widths along the x and y directions in the source plane, respectively, $\theta$ represents the angle that the phase has been rotated in the Cartesian coordinate.

The new phase in Equation (7) becomes separable, which we called the twisting phase. The twisting phase can act on completely coherent beams and is convenient to be generated by spatial light modulator (SLM) in experiments. We noticed that the form of the twisting phase is similar to the $Z_5$ in the Zernike polynomial, which represents the oblique astigmatism [56]. It is easy to find that the phase of the cylindrical lens placed at an angle of 45° can be regarded as a superposition of the circular lens and the twisting phase,

$$\begin{aligned} \psi_{rcl} &= -i\frac{k}{2f}(x\cos 45° - y\sin 45°)^2 = -i\frac{k}{4f}(x^2 + y^2 - 2xy) \\ &= -i\frac{k}{4f}(x^2 + y^2) - (-i\frac{k}{2f}xy) = \psi_{lens} + (i\mu'xy), \end{aligned} \tag{8}$$

where $\psi_{rcl}$ is the phase of a cylindrical lens with focal length $f$ placed at an angle of 45°, $\psi_{lens}$ is the phase of a lens with focal length $2f$, $\mu' = \frac{k}{2f}$. Equation (8) infers that the twisting phase has a certain relationship with the twisted beams produced by the cylindrical lens group.

### 3. Properties of Various Types of Beams That Carry Twist Phase

The transmission characteristics of several types of beams that carry twist phases will be introduced in this part. The Fresnel diffraction formula can be used to analyze the radiation field generated by partially coherent sources [54]

$$W(\boldsymbol{\rho}_1, \boldsymbol{\rho}_2, z) = \iint W_0(\boldsymbol{r}_1, \boldsymbol{r}_2) G_z^*(\boldsymbol{\rho}_1, \boldsymbol{r}_1, z) G_z(\boldsymbol{\rho}_2, \boldsymbol{r}_2, z) \mathrm{d}^2 \boldsymbol{r}_1 \mathrm{d}^2 \boldsymbol{r}_2, \tag{9}$$

where $\boldsymbol{\rho}_1$ and $\boldsymbol{\rho}_2$ are arbitrary transverse position vectors in observed plane; $\boldsymbol{r}_1$ and $\boldsymbol{r}_2$ are arbitrary transverse position vectors in source plane; $G_z$ is a free-space propagator, whose form in the paraxial approximation is shown as

$$G_z(\boldsymbol{\rho}, \boldsymbol{r}, z) = -\frac{ik}{2\pi z} \exp(ikz) \exp[\frac{ik}{2z}(\boldsymbol{\rho} - \boldsymbol{r})^2]. \tag{10}$$

where $k$ represents the wave number and $z$ is the transmission distance. With the help of Equations (9) and (10), the CSD function of the partially coherent beams can be derived, which could be used to analyze the transmission characteristics, such as spectral density $S$, DOC $\mu$ and degree of polarization (DOP) $P$,

$$S(\boldsymbol{\rho}, z) = W(\boldsymbol{\rho}, \boldsymbol{\rho}, z), \tag{11}$$

$$\mu(\boldsymbol{\rho}_1, \boldsymbol{\rho}_2, z) = \frac{W(\boldsymbol{\rho}_1, \boldsymbol{\rho}_2, z)}{\sqrt{S(\boldsymbol{\rho}_1, z) S(\boldsymbol{\rho}_2, z)}}, \tag{12}$$

$$P(\boldsymbol{\rho}, z) = \frac{|W_{xx}(\boldsymbol{\rho}, \boldsymbol{\rho}, z) - W_{yy}(\boldsymbol{\rho}, \boldsymbol{\rho}, z)|}{W_{xx}(\boldsymbol{\rho}, \boldsymbol{\rho}, z) + W_{yy}(\boldsymbol{\rho}, \boldsymbol{\rho}, z)} \tag{13}$$

#### 3.1. Twisted Gaussian Schell-Model Array (TGSMA) Beams

The CSD function of a TGSMA beam in the source plane is set as [23,57–59]

$$
\begin{aligned}
W_0(\boldsymbol{r}_1, \boldsymbol{r}_2) = \quad & \exp(-\frac{x_1^2 + x_2^2}{4\sigma_x^2}) \exp(-\frac{y_1^2 + y_2^2}{4\sigma_y^2}) \exp[-\frac{(x_1 - x_2)^2}{2\delta_x^2}] \exp[-\frac{(y_1 - y_2)^2}{2\delta_y^2}] \\
\times & \sum_{n_x=-P}^{P} \cos[C_x(x_1 - x_2)] \sum_{n_y=-Q}^{Q} \cos[C_y(y_1 - y_2)] \exp[-iu(x_1 y_2 - x_2 y_1)],
\end{aligned} \tag{14}
$$

where $P = (N_x - 1)/2$ and $Q = (N_y - 1)/2$; $N_x$ and $N_y$ are positive integrals determining the number of lobes of the array; $C_j = 2\pi n_j R_j / \delta_j$ and $R_j$ are parameters about coherence.

Substituting from Equations (10) and (14) into Equation (9), the CSD function of the TGSMA beam in transmission is calculated [23]. The spectral density and the DOC of the TGSMA beam propagating in free space are shown in Figure 1, where the parameters are chosen as $N_x = N_y = 3$, $\sigma_x = \delta_x = 1$ mm, $\sigma_y = \delta_y = 0.3$ mm, $R_x = 2R_y = 3$ mm, $u = 5$ mm$^{-2}$ and $\lambda = 632.8$ nm. It is obvious that the spectral density of the TGSMA beam will split from an elliptical spot to an array and rotate clockwise during propagation, whereas its DOC could merge and rotate in the opposite direction.

#### 3.2. Twisted Rectangular Multi-Gaussian Schell-Model (TRMGSM) Beam

We studied the propagation characteristics of the TRMGSM beam in free space and ocean turbulence, whose CSD function in the source plane is shown as [10,37]

$$
\begin{aligned}
W_0(\boldsymbol{r}_1, \boldsymbol{r}_2) = \quad & \sum_{m_1=1}^{M_1} \binom{M_1}{m_1} \frac{(-1)^{m_1-1}}{\sqrt{m_1}} \sum_{m_2=1}^{M_2} \binom{M_2}{m_2} \frac{(-1)^{m_2-1}}{\sqrt{m_2}} \exp[-(\sigma - \frac{\alpha u^2}{2m_1})(y_1^2 + y_2^2) - \frac{(x_1 - x_2)^2}{4m_1 \alpha} - \frac{\alpha u^2 (y_1 - y_2)^2}{4m_1}] \\
\times & \exp[-(\sigma - \frac{\beta u^2}{2m_2})(x_1^2 + x_2^2) - \frac{(y_1 - y_2)^2}{4m_2 \beta} - \frac{\alpha u^2 (x_1 - x_2)^2}{4m_2}] \exp[-\frac{iu(x_1 + x_2)(y_1 - y_2)}{2m_2} - \frac{iu(x_1 - x_2)(y_1 + y_2)}{2m_1}].
\end{aligned} \tag{15}
$$

Substituting from Equations (10) and (15) into Equation (9), and introducing additional correction terms about the power spectral of ocean turbulence $\Phi_n(\kappa)$, the CSD function in transmission is derived [37,60,61].

Figure 2 shows the spectral density and DOC generated by a TRMGSM source at several propagation distances in free space and oceanic turbulence with the following parameters: $\sigma = 11$ mm$^{-2}$, $\alpha = \beta = 0.05$ mm$^2$, $M_1 = M_2 = 5$, $u = 10$ mm$^{-2}$ and $\lambda = 630$ nm. It is found that the rectangular beam spot shows four rotated peaks. The petal-like structure of DOC also rotates and contracts in transmission. The oceanic turbulence does not affect the distribution and rotation during transmission, it only transforms the spectral density and DOC into Gaussian distribution in the far field.

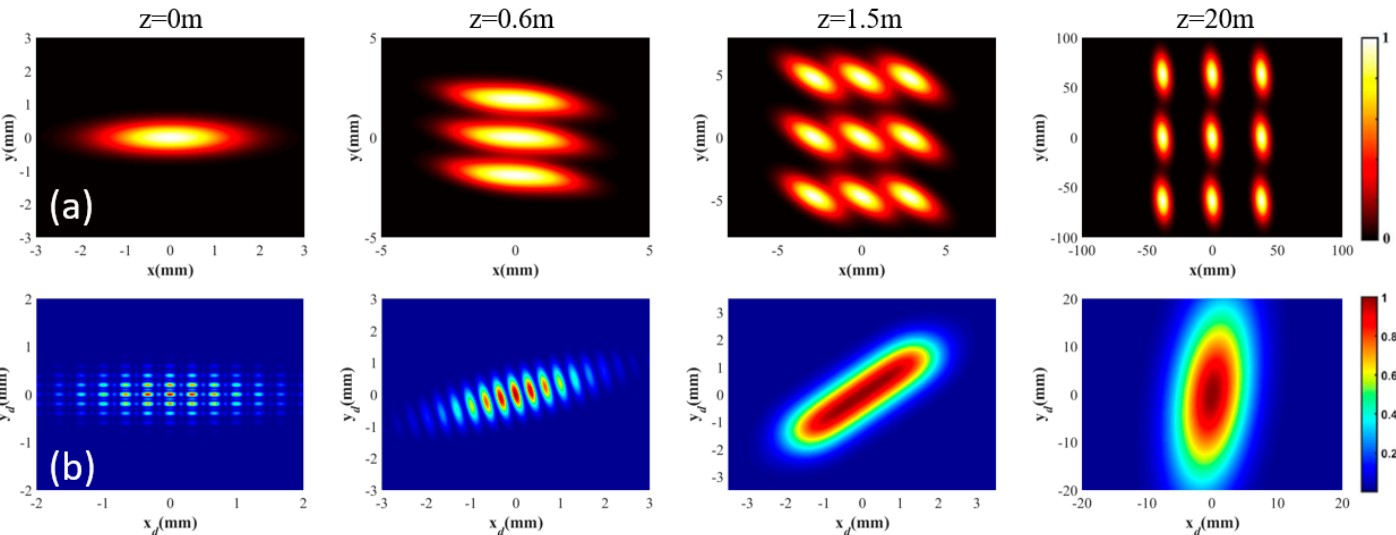

**Figure 1.** The distribution of (**a**) spectral density and (**b**) degree of coherence (DOC) associated with a typical twisted Gaussian Schell-model (TGSMA) beam propagating in free-space.

### 3.3. Electromagnetic Twisted Gaussian Schell-Model Array (EM TGSMA) Beam

As a kind of vectorial beam, the CSD of EM TGSMA beams is described by a $2 \times 2$ matrix instead, whose element is set as [26,62]

$$
W_{ij}(\boldsymbol{r}_1, \boldsymbol{r}_2) = \exp\left(-\frac{x_1^2 + x_2^2}{4\sigma_{1ij}^2}\right) \exp\left(-\frac{y_1^2 + y_2^2}{4\sigma_{2ij}^2}\right) \exp\left[-\frac{(x_1 - x_2)^2}{2\delta_{1ij}^2}\right] \exp\left[-\frac{(y_1 - y_2)^2}{2\delta_{2ij}^2}\right]
$$
$$
\times \sum_{n_1=-P}^{P} \cos[C_{1ij}(x_1 - x_2)] \sum_{n_2=-Q}^{Q} \cos[C_{2ij}(y_1 - y_2)] \exp[-iu_{ij}(x_1 y_2 - x_2 y_1)]. \tag{16}
$$

where subscript $i, j = x, y$ represents the position in CSD matrix. Substituting from Equation (16) into Equation (9), the CSD matrix of the EM TGSMA beam is calculated to study the transmission characteristics [26]. The spectral density, DOC and DOP of the EM TGSMA beam with parameters $R_1 = 2R_2 = 3$ mm, $N_1 = 2N_2 = 2$ and $\lambda = 632.8$ nm are shown in Figure 3. The spectral density of EM TGSMA beam will split during transmission. The distribution of DOC of the beam has a compression process in the x direction. By changing the proportional relationship between the parameters, the direction of rotation of the DOC and DOP could be controlled.

### 3.4. Beams with High-Order Twist Phase

A new family of partially coherent beams incorporating a series of non-separable phases are introduced [24]. According to the special weight function and kernel function, the CSD function in the source plane is derived as

$$
W_0(\boldsymbol{r}_1, \boldsymbol{r}_2) = \exp\left(-\frac{r_1^2 + r_2^2}{4\sigma^2}\right) \exp\left[-\frac{(x_1^n - x_2^n)^2 + (y_1^n - y_2^n)^2}{2\delta^{2n}}\right] \exp[-iu(x_1^n y_2^n - x_2^n y_1^n)], \tag{17}
$$

with $\frac{1}{2\delta^{2n}} = \frac{4\alpha^2 + \gamma^2}{4\beta}$; $u = \frac{2\alpha\gamma}{\beta}$.

Due to the special phase space structure, this family of beams also have unusual orbital angular momentum (OAM) characteristics [24,63]. The beams with $n = 2$ are used as examples to illustrate the evolution of the spectral density of beams in transmission, which is called the stretched non-uniform model (SNUM) beams. Different to obtaining an analytical formula, the spectral density is derived by fast Fourier transform (FFT) algorithm in MATLAB [64]. It could be concluded that for an SNUM beam with the quartic phase, t0e three-dimensional structure exhibits a wing-like shape after stretching, which is the result of the interaction between the magnitude of the source coherence state and the high-order twist phase [65]. By comparing the SNUM beams with close $u$, the process of unfolding wings in the three-dimensional structure is more clearly demonstrated.

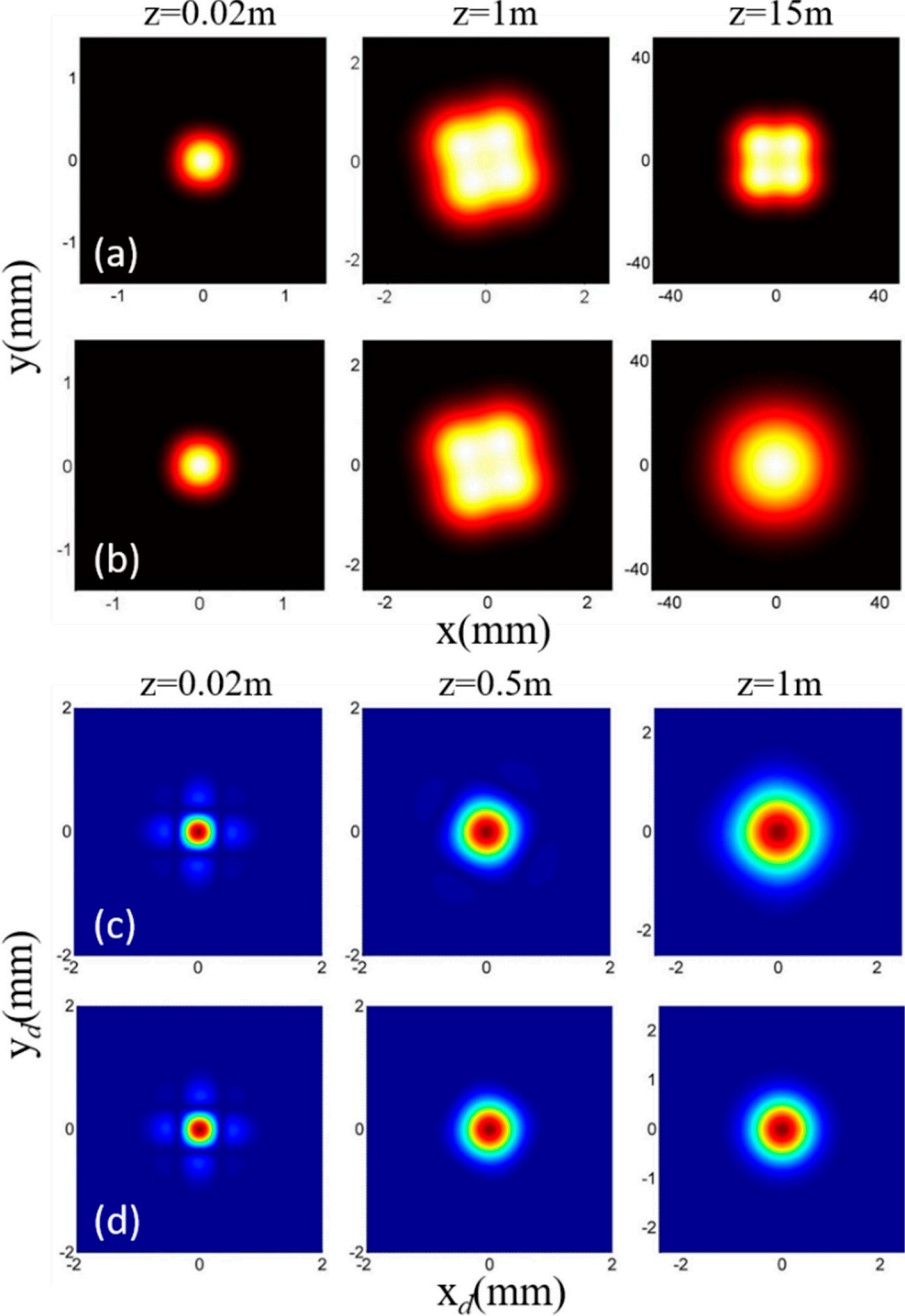

**Figure 2.** The (**a**,**b**) spectral density and (c, d) DOC generated by a Twisted Rectangular Multi-Gaussian Schell-Model (TRMGSM) source in (**a**,**c**) free space and (**b**,**d**) oceanic turbulence during propagation.

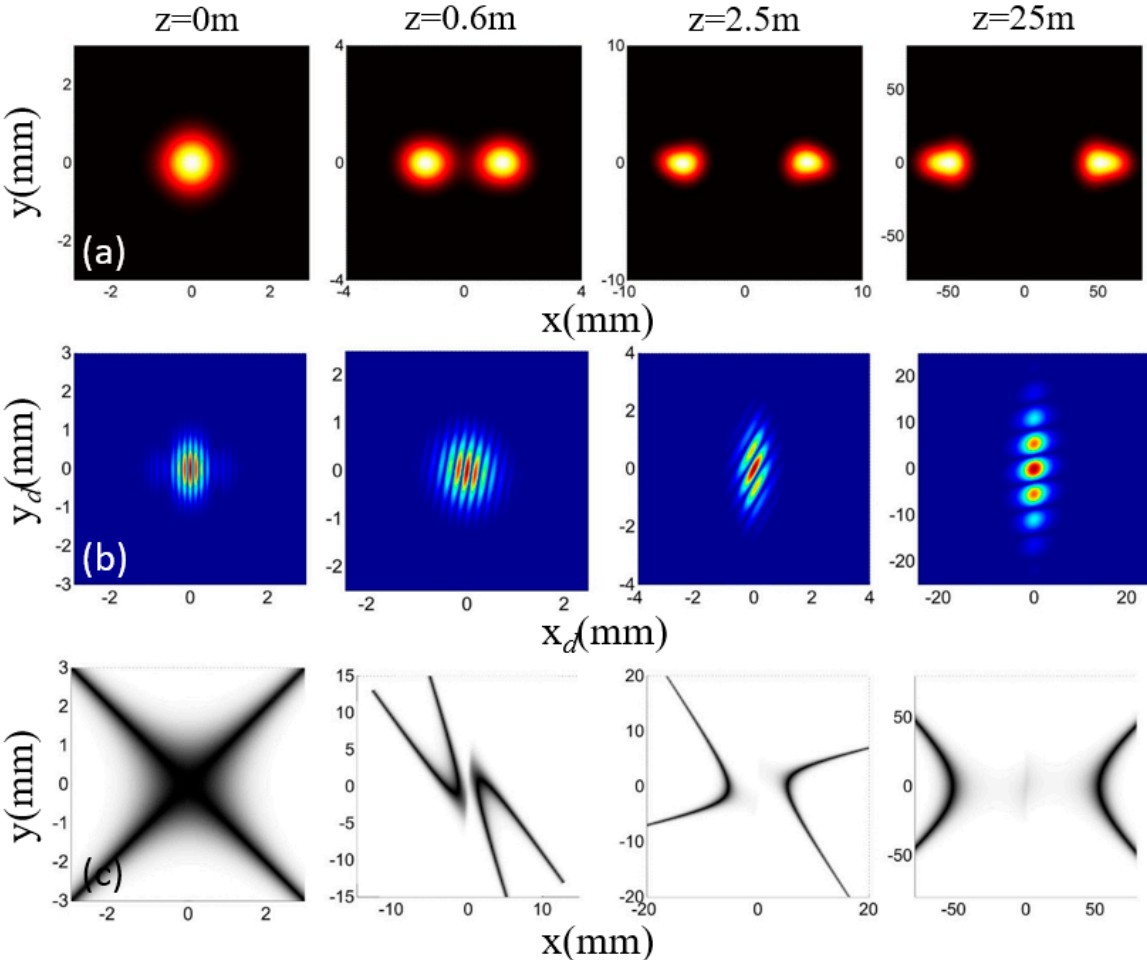

**Figure 3.** The (**a**) spectral density, (**b**) DOC and (**c**) degree of polarization (DOP) generated by an EM TGSMA source during propagation.

## 4. Effect of the Twisting Phase on Beams

The twisting phase is also a kind of phase that can rotate the spectral density during transmission. Compared with the traditional twist phase, the twisting phase is separable and could exist in completely coherent beams. Its equivalent phase form in two-dimensional plane is $iu(x\cos\theta - y\sin\theta)(y\cos\theta + x\sin\theta)$ [43]. Figure 4 shows phase graphs of the twisting phase that were painted in MATLAB, indicating that the twisting phase can be easily added into completely coherent beams or partially coherent beams by SLM [50,51].

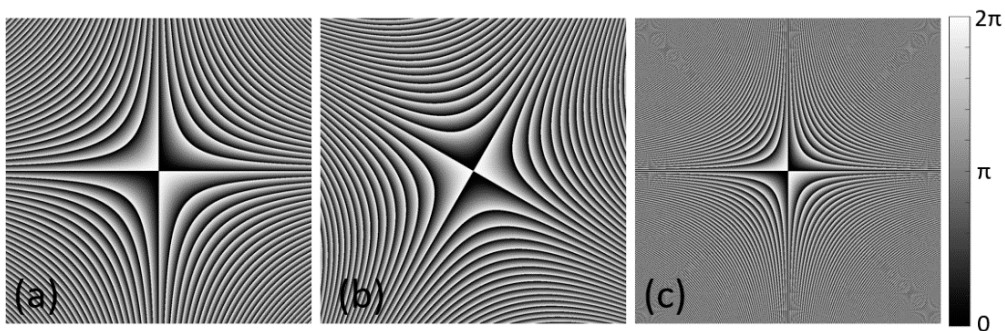

**Figure 4.** Phase graphs of the twisting phase. (**a**) $u = 10$ mm$^{-2}$, $\theta = 0$; (**b**) $u = 10$ mm$^{-2}$, $\theta = \pi/3$; (**c**) $u = 30$ mm$^{-2}$, $\theta = 0$.

In this part, we mainly introduce the relevant study progress of the twisting phase in the beam, including the effect of this phase in completely coherent beams, partially coherent beams, and applications in particle manipulation.

### 4.1. The Twisting Phase in Completely Coherent Beams

Consider the anisotropic vortex beams carrying on the twisting phase, whose electric field in the source plane is expressed as [50,51]

$$E_0(x,y) = \exp(-\frac{x^2}{4\sigma_x^2})\exp(-\frac{y^2}{4\sigma_y^2})\exp(im\varphi)\exp(-iuxy), \tag{18}$$

where $m$ represents the topological charge; $\varphi$ is the argument corresponding to each coordinate. When $m$ equals 0, the beam degenerates into the anisotropic Gaussian beam. The spectral density could be calculated by Fresnel diffraction formula or FFT algorithm in MATLAB. The spectral density of the anisotropic Gaussian beam and vortex beam under the effect of the twisting phase during propagation are shown in Figure 5, where the parameters are $\sigma_x = 2\sigma_y = 1$ mm and $u = 30$ mm$^{-2}$. The anisotropic completely coherent beam with the twisting phase also exhibits a rotation during transmission, which is similar to the TGSM beam. By changing the $\theta$ in the phase, the angle of rotation could be controlled [50].

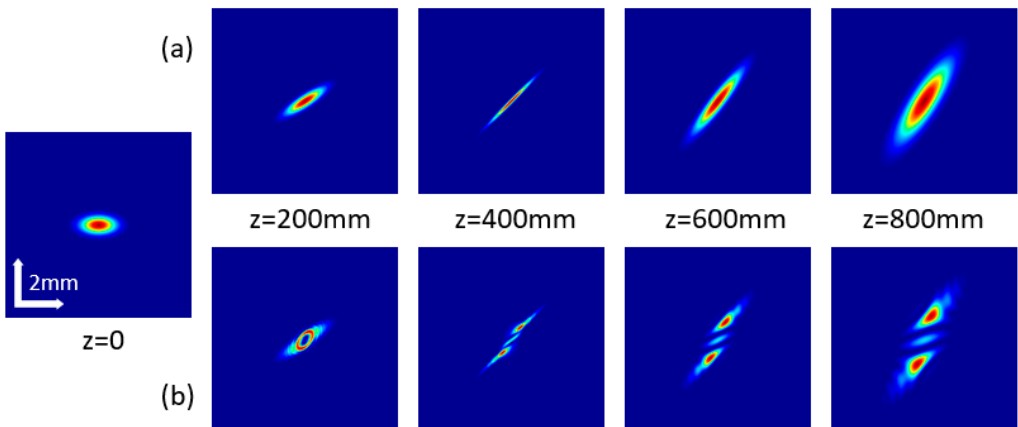

**Figure 5.** The spectral density of the (**a**) anisotropic Gaussian beam and (**b**) vortex beam ($m = 2$) under the effect of the twisting phase during propagation.

It is worth noting that under the effect of the twisting phase, the spectral density of the vortex beam exhibits striped distribution. The spectral density of vortex beams carrying on the twisting phase with different topological charges at $z = 600$ mm is shown in Figure 6. It could be concluded that the stripe in spectral density is related to the topological charge of the beam. The number of dark stripes is equal to the absolute value of the topological charge, and the direction of the stripes is controlled by the sign of the topological charge, meaning that the twisting phase could be used to measure the topological charge conveniently [51,66].

### 4.2. The Twisting Phase in Partially Coherent Beams

We also studied the role of the twisting phase in partially coherent beams. The rotating anisotropic Gaussian Schell-model (RAGSM) beam is introduced, whose CSD function in source plane is expressed as [43,52]

$$\begin{aligned}W_0(\boldsymbol{r}_1,\boldsymbol{r}_2) = \quad & \exp(-\frac{x_1^2+x_2^2}{4\sigma_x^2})\exp(-\frac{y_1^2+y_2^2}{4\sigma_y^2})\exp[-\frac{(x_1-x_2)^2}{2\delta_x^2}]\exp[-\frac{(y_1-y_2)^2}{2\delta_y^2}] \\ & \times \exp\{-iu[(x_1\cos\theta - y_1\sin\theta)(x_1\sin\theta + y_1\cos\theta) \\ & \quad -(x_2\cos\theta - y_2\sin\theta)(x_2\sin\theta + y_2\cos\theta)]\}.\end{aligned} \tag{19}$$

The total average OAM per photon of the RAGSM beam is derived as

$$L_{orbit} = u\hbar(\sigma_x^2 - \sigma_y^2)\cos 2\theta. \tag{20}$$

It can be concluded from Equation (20) that the OAM carried by the beam is independent of the coherent source, and the rotation only appears in anisotropic beams with $\cos 2\theta \neq 0$.

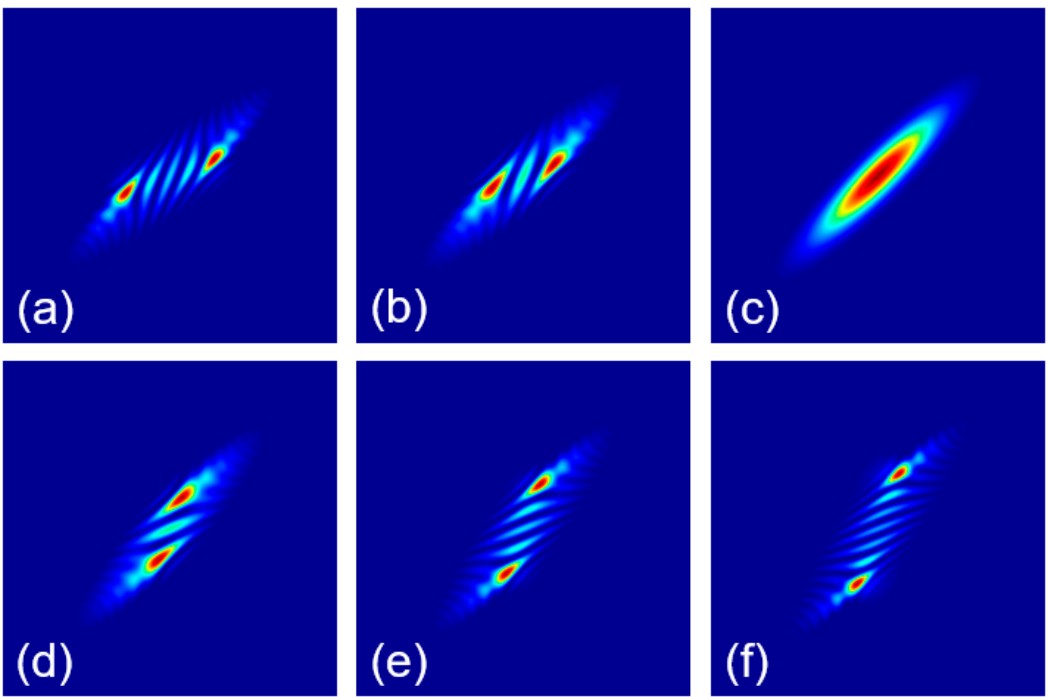

**Figure 6.** The spectral density of vortex beams under the effect of the twisting phase at $z = 600$ mm. The topological charge $m$ equals (**a**) $-4$, (**b**) $-2$, (**c**) 0, (**d**) 2, (**e**) 4, (**f**) 6.

The spectral density and DOC of partially coherent beams are still calculated by the CSD function from Fresnel diffraction formula. The parameters of the beam are chosen as $\sigma_x = 2\sigma_y = 1$ mm, $5\delta_x = \delta_y = 0.1$ mm and $\lambda = 532.8$ nm. In order to analyze the rotation better, an additional lens with $f = 200$ mm is placed at the source plane. The rotation angle of spectral density and DOC of the RAGSM beam during transmission are shown in Figure 7. The spectral density and DOC of the RAGSM beam rotate during transmission, even back to the original angle, which is called reversal rotation. Whether reversal rotation can occur depends on the twist strength coefficient $u$, the spectral density, and DOC distribution of beam source [52].

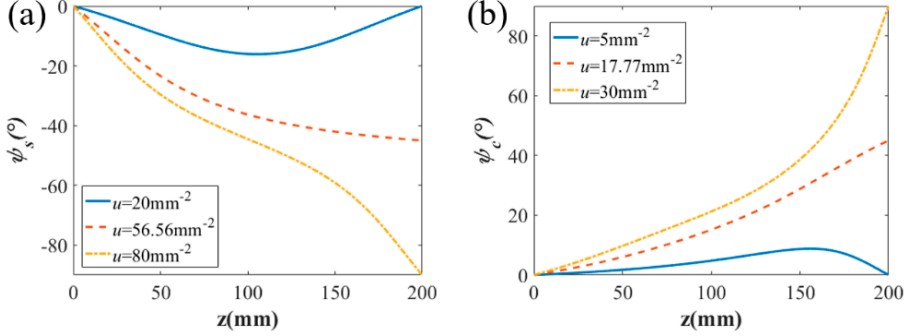

**Figure 7.** The rotation angle of (**a**) spectral density and (**b**) DOC of rotating anisotropic Gaussian Schell-model (RAGSM) beams with different $u$.

### 4.3. The Twisting Phase in Optical Trapping Rayleigh Particles

We investigated the focusing properties and the radiation forces of the rotating anisotropic generalized multi-Gaussian Schell model (RAGMGSM) beam, whose CSD function is shown as [53,67]

$$
\begin{aligned}
W_0(\boldsymbol{r}_1, \boldsymbol{r}_2) = \quad & I_0 \exp\left(-\frac{x_1^2+x_2^2}{4\sigma_x^2}\right)\exp\left(-\frac{y_1^2+y_2^2}{4\sigma_y^2}\right)\frac{1}{C_0}\sum_{n=1}^{2M}\sum_{m=1}^{2M}(-1)^{n+m}A_{nm}B_{nm} \\
& \times \exp\left[-\frac{B_{nm}(x_1-x_2)^2}{2\delta_x^2}\right]\exp\left[-\frac{B_{nm}(y_1-y_2)^2}{2\delta_y^2}\right] \\
& \times \exp\{-iu[(x_1\cos\theta - y_1\sin\theta)(x_1\sin\theta + y_1\cos\theta) \\
& \quad - (x_2\cos\theta - y_2\sin\theta)(x_2\sin\theta + y_2\cos\theta)]\},
\end{aligned}
\tag{21}
$$

where $I_0$ is a constant about input power and refractive index of the medium; $A_{nm} = \begin{pmatrix} 4M \\ 2n-1 \end{pmatrix}\begin{pmatrix} 4M \\ 2m-1 \end{pmatrix}$; $B_{nm} = \frac{2nm}{n+m}$; $C_0$ is the normalization factor.

The focused RAGMGSM beam produces radiation force on Rayleigh dielectric particles. According to the Rayleigh scattering model, radiation force consists of scattering force and gradient force. In this case, the particles can be regarded as a point dipole in the field. The scattering and gradient force can be derived by [68]

$$
\begin{aligned}
\overrightarrow{F}_{Scat} &= \overrightarrow{e_z}\frac{8\pi n_m k^4 a^6}{3c}\left(\frac{\eta^2-1}{\eta^2+2}\right)^2 I_{out}, \\
\overrightarrow{F}_{Grad} &= \frac{2\pi n_m a^3}{c}\left(\frac{\eta^2-1}{\eta^2+2}\right)\nabla I_{out},
\end{aligned}
\tag{22}
$$

where $\eta$ represents the relative refractive index and $n_m$ denotes the refractive index of the medium; $a$ is the radius of the Rayleigh particle.

The radiation force produced by RAGMGSM beam with different $u$ on the Rayleigh particle is shown in Figure 8, with parameters are $P = 1$ W, $a = 30$ nm, $n_m = 1.33$, $\eta = 0.75$, $\theta = 60°$ and $\lambda = 632.8$ nm. Figure 8a depicts the radiation force of RAGMGSM beams with different $u$ at the equilibrium position. It could be seen that the radiation force becomes weaker when the twisted strength coefficient $u$ increases, meaning that the trapping stability and the stable trapping region decrease. Figure 8b shows the two-dimensional radiation force distribution near the trapped plane ($\Delta z = 0.232$ µm). It could be noticed that the direction of the transverse gradient force always points to the center; the longitudinal gradient force could push the Rayleigh particles to the equilibrium point, which is indicated by a red circle; the scattering force is always in the same direction as the beam propagation.

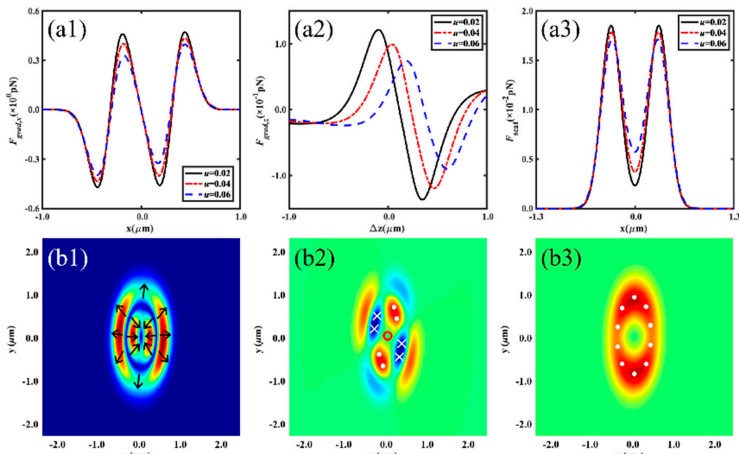

**Figure 8.** (**a**) The radiation force produced by rotating anisotropic generalized multi-Gaussian Schell model (RAGMGSM) beam with different $u$ on the Rayleigh particle; (**b**) The distribution of radiation force near the trapped plane with $u = 0.04$ mm$^{-2}$.

## 5. Summary and Prospects

We briefly introduced the latest developments in our group on beams carrying on traditional twist phase or twisting phase. The special properties of optical twist shown in these studies have direct or potential applications in many fields, such as optical conveying, optical communication, three-dimensional laser material processing, topological charge measurement, particle capture and manipulation. The general results in this paper deepen the understanding of optical twist and provide a new perspective of related research. Further studies in this field may focus on groups designing beams with special twisted structures theoretically and experimentally, and discovering new characteristics and applications. We believe that this field will develop rapidly, revealing more interesting phenomenon and valuable applications.

**Author Contributions:** Conceptualization, L.W. and Z.L.; methodology, L.W. and Z.L.; investigation, Z.L., L.W., Y.Z. (Yujie Zhou) and Y.Z. (Yao Zhang); writing—original draft preparation, Z.L.; writing—review and editing, Z.L. and D.Z.; supervision, D.Z.; project administration, D.Z. funding acquisition, D.Z. All authors have read and agreed to the published version of the manuscript.

**Funding:** This research was funded by National Natural Science Foundation of China, grant number 11874321.

**Institutional Review Board Statement:** Not applicable.

**Informed Consent Statement:** Not applicable.

**Data Availability Statement:** No new data were created or analyzed in this study. Data sharing is not applicable to this article.

**Conflicts of Interest:** The authors declare no conflict of interest.

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
