# Peer review of "Progress on Studies of Beams Carrying Twist"

_photonics, doi:10.3390/photonics8040092_

Round 1

Reviewer 1 Report

The author presents a review paper on partially coherent beams with twist. In particular, their review includes papers dealing with both the twist phase introduced by Simon and Mukunda in 1993, and the one introduced by the authors themselves, and that they refer to as "the new twisting phase".

I think that the paper could be useful and deserves publication.

Nonetheless, the following points should be considered.

• The bibliography could be more complete. For example, at least the following papers on the subject should be quoted:

- Gori, F., Bagini, V., Santarsiero, M., Frezza, F., Schettini, G., Schirripa Spagnolo, G. (1994). Coherent and partially coherent twisting beams. Optical Review, 1(2), 143–145.

- Ponomarenko, S. (2001). A class of partially coherent beams carrying optical vortices. Journal of the Optical Society of America A - Optics Image Science and Vision, 18(1), 150–156.

- Hyde, M.W. (2020). Twisted space-frequency and space-time partially coherent beams. Sci Rep 10, 12443 

• The angle $\theta$ in Eq.(7) has not been defined.

• The authors states that the "new twisting phase" has a certain relationship with the twisted beams produced by a cylindrical lens group. Indeed, it seems that it can be reduced to the effect of an astigmatic lens having focal lengths $f$ and $-f$ along two orthogonal axes, possibly rotated by a certain angle with respects to the $x$ and $y$ axes. The authors should clarify such relationship. Furthermore, if the above interpretation is correct, it seems that the use of SLMs could not be necessary for the synthesis of such phase. 

• Except for the case $m=0$, I suspect that Eq.(17) is not correct. In fact, if $m$ differs from 0, the phase is not defined at the axes origin and the field should vanish there.

Author Response

Response: We thank the reviewer for the comments and suggestions. We will respond to each comment.

  1. The reviewer's suggestion about bibliography is very valuable and we have added citations to these papers.
  2. in Equation (7) has been defined as the angle that the phase has been rotated in the Cartesian coordinate.
  3. In the second part of the paper, we added a description of the relationship between the new twisting phase and the lens. It can be calculated that the phase of a cylindrical lens rotated 45 degrees can be regarded as the superposition of a lens phase and a new twisting phase, which is helpful to understanding the relationship between the new twisting phase and the lens group.
  4. About Equation 18 (original Equation 17), we do not agree with the reviewer. Equation 18 is the electric field at the source plane of an anisotropic Gaussian beam carrying the vortex phase and the new twisting phase. When m is not equal to 0, the phase of the beam at the origin is uncertain, which is a characteristic of vortex beams, but it does not conflict with the addition of a new twisting phase. When z=0, the electric field of the source plane is directly multiplied and the phase part disappears completely due to conjugate, the spectral density shows elliptical distribution. Then, the center of the beam appears a dark spot with a ring-shaped distribution, which is due to the uncertain phase brought by the vortex beam. At a farther distance, the new twisting phase splits the topological charge, and the spectral density is distributed in stripes. These changes of spectral density are consistent with them in Figure 5(b). Therefore, Equation 18 is correct.

Thanks again for the reviewer’s suggestion.

Reviewer 2 Report

This paper describes beams carrying twist and new twisting phase. This paper provides interests concerning unique phase for readers. I recommend to accept for the publication in Photonics. However, I think this paper needs to explain all characters in equations

Author Response

Response: We thank the reviewer for the comments and suggestions. We have explained the characters in equations in this paper to make the equations easier to be understand.

Reviewer 3 Report

In this brief review, the authors consider rotating partially coherent beams with a twist phase.

TGSM beams (4) from [1] have a twist phase, which can only be present in partially coherent fields. Further, the authors consider beams with a “new twisting phase” from [41]. This phase can also be present in coherent light fields [42, 43]. But the “new twisting phase” is an astigmatic phase that has long been known in coherent optics.

For the first time, an astigmatic transformation based on a rotated cylindrical lens was used in a mode converter [Opt. Commun. 83, 123-135 (1991)].

In 2004, based on the astigmatic phase xy, Laguerre-Hermite-Gauss beams were obtained [J. Opt. A: Pure Appl Opt 6 (5), S157-S161 (2004)].

Orbital angular momentum (19) was obtained back in 1997 in [Opt. Commun. 144, 210-213 (1997)].

Other works on astigmatic beams are also known, which were done before the works of the authors of 2019-2020 [42-44].

For example, Appl Opt 56 (14) 4095-4104 (2017); Opt Express 26 (1) 141-156 (2018).

Note that the astigmatic phase not only has the form xy, but also x ^ 2-y ^ 2. This twist phase could also be considered in the review. For completeness of consideration of this problem, the authors should include the above works in the review. After taking into account my comments, this review can be published.

Author Response

Response: We thank the reviewer for the comments and suggestions. The reviewer's suggestion about bibliography is very significant. We have added citations to these papers to display the research about the new twisting phase better.

Round 2

Reviewer 3 Report

The authors took into account all my comments and the work can be published

Author Response

Thanks!